# Ancient and methane-derived carbon subsidizes contemporary food webs

Amanda G. DelVecchia[1,†], Jack A. Stanford[1,†] & Xiaomei Xu[2]

While most global productivity is driven by modern photosynthesis, river ecosystems are supplied by locally fixed and imported carbon that spans a range of ages. Alluvial aquifers of gravel-bedded river floodplains present a conundrum: despite no possibility for photosynthesis in groundwater and extreme paucity of labile organic carbon, they support diverse and abundant large-bodied consumers (stoneflies, Insecta: Plecoptera). Here we show that up to a majority of the biomass carbon composition of these top consumers in four floodplain aquifers of Montana and Washington is methane-derived. The methane carbon ranges in age from modern to up to >50,000 years old and is mostly derived from biogenic sources, although a thermogenic contribution could not be excluded. We document one of the most expansive ecosystems to contain site-wide macroinvertebrate biomass comprised of methane-derived carbon and thereby advance contemporary understanding of basal resources supporting riverine productivity.

[1] Flathead Lake Biological Station, 32125 Bio Station Lane, Polson, Montana 59860, USA. [2] University of California Irvine, Earth System Science, 2222 Croul Hall, Irvine, California 92697-3100, USA. † Present address: Allegheny College, Department of Biology, B300 Steffee Hall, 520 North Main St., Meadville, Pennsylvania 16335, USA (A.G.D.); 36 Tuckaway Road, Twisp, Washington 98856, USA (J.A.S.). Correspondence and requests for materials should be addressed to A.G.D. (email: amanda.delvecchia@gmail.com).

Two landmark papers in 1974 and 1988 revolutionized our view of river systems[1,2]. These works demonstrated that the shallow alluvial aquifers of river floodplains were abundantly populated by diverse large-bodied hyporheic stoneflies (Insecta:Plecoptera) that spent their nymphal stages entirely underground before emerging from the river channel as flying adults. The papers highlighted the broad extent of surface and groundwater interchange, and additionally underscored the importance of hydrologic and biogeochemical connectivity for maintaining biodiversity and productivity of river ecosystems. In the decades that followed, knowledge of the importance of riverine aquifers has expanded but the question has persisted: how do these abundant large-bodied consumers survive in the highly oligotrophic, dark and carbon-limited environment of the aquifer?

River floodplains worldwide are underlain by shallow alluvial aquifers where interstitial flow is driven by penetration of river water into the bed sediments. In gravel-bed systems these aquifers are extremely porous and generally well-oxygenated. The aquifers may contain diverse and abundant meiofauna as well as large-bodied stoneflies (Supplementary Fig. 1)[1–3]. The presence of speciose communities is a conundrum because productivity is generally limited by labile organic carbon availability and microbial productivity is extremely low[3–5]. The Nyack Floodplain on the Middle Fork of the Flathead River in northwestern Montana (Fig. 1) provides a well-documented example of an expansive alluvial aquifer that is ultra-oligotrophic yet paradoxically supports a diverse and productive food web with large (up to 3 cm length) stonefly larvae as top consumers.

The Nyack aquifer is contained in gravel and cobble bed-sediments that were deposited during the last glacial retreat $\sim$7,000–10,000 years ago[6,7] and subsequently reworked by cut and fill alluviation associated with river flooding[6]. The 20–50+ m thick aquifer is occluded by Precambrian bedrock overlain by glacial outwash clays and a Tertiary shale (Kishenehn) formation that is carboniferous and thus a potential source of thermogenic carbon[8]. The aquifer is characterized by extreme hydraulic conductance up to 11.6 cm s$^{-1}$ (ref. 9) and is exclusively recharged by the river[6]. It is therefore considered a voluminous 'hyporheic' zone where surface and ground waters interchange (Supplementary Fig. 2). Water residence times vary from hours to three years in relation to lengths of flow paths from the river through the aquifer[9]. Overall, the aquifer is well oxygenated because oxygen diffuses from the vadose zone of the floodplain[10]. Dissolved organic carbon concentrations are consistently <2 mg l$^{-1}$ and microbial productivity is ultra-limited by paucity of labile organic carbon[4]. Along short flow paths near Along short flow paths near the river (for example, through gravel bars), respiration of allochthonous carbon causes a predictable drop in dissolved oxygen (DO) and dissolved organic carbon (DOC)[11]. However, along longer flow paths through the entire aquifer, an anomalous increase in organic carbon lability occurs, suggesting carbon fixation perhaps by chemoautotrophy and/or methanotrophy[11]. The occurrence of chemoautotrophy and/or methanotrophy in the aquifer also has been proposed as a solution to the documented imbalance in the Nyack aquifer carbon budget[12].

Thus, we investigated the source and role of methane as a potential subsidy to floodplain aquifer food webs, mainly at Nyack but also at three other locations: the Kalispell floodplain on the main stem of the Flathead River in Northwest Montana, the Jocko River floodplain in Southwest Montana and the Methow River floodplain in Washington. At each of these sites a grid of slotted, but not screened, groundwater monitoring wells was available for sampling. Of this suite of aquifers only the Nyack is underlain by a hydrocarbon-containing shale formation.

We posited: (1) what is the source of the methane, (2) what are the contributions of various methane sources to stonefly biomass and (3) is a methane subsidy in alluvial aquifers a widespread phenomenon? To identify methane sources, we measured the carbon and deuterium stable isotope ratios of dissolved methane, the radiocarbon ages of dissolved methane, and methane, ethane and propane concentrations. In order to understand the contributions of various methane-derived carbon sources to biomass (question 2), we measured carbon stable isotope ratios and radiocarbon ages of stonefly biomass and organic matter, and then incorporated these values into Bayesian mixing models that were parameterized using a suite of scenarios to give a range of reasonable and conservative estimates of source contributions to biomass. We addressed question 3 by comparing results among study sites.

## Results

**Methane sources in the nyack aquifer**. At Nyack, we collected samples from seven wells (Fig. 1) previously shown to contain the full suite of aquifer biota (Supplementary Fig. 1). One of the wells had a residence time of 45 days while all others ranged from 117 to 305 days (Supplementary Table 1). We sampled at two depths, 1 and 4 m below the base-flow water table, approximately every three weeks from August 2013 to August 2015. In addition, we collected samples near the bottom of well HA10 specifically to target potential shale off-gassing of methane because this well had high methane concentrations deeper in the well. Only three wells, HA10, HA12 and HA17, yielded methane concentrations >1 µmol l$^{-1}$ (Supplementary Fig. 3A). These three were the only wells with methane concentrations high enough for us to measure stable isotope values. Only wells HA10 and HA12 occasionally had high enough concentrations to measure radiocarbon. In these two wells, maximum concentration reached 10% saturation (Supplementary Fig. 3B).

In wells HA10, HA12 and HA17, we compared our measured methane stable isotope ratios to known characterizations of methane sources based on the ratios of deuterium and carbon stable isotope signatures (Fig. 2a)[13–17]. While methanogenic signatures typically are easy to identify as having light (low) carbon and hydrogen isotopic ratios, any deviation from these combinations leads to uncertainty as to whether the resulting heaviness might be caused by variable sources, such as a thermogenic methane contribution, or by microbial oxidation of biogenic methane, which leaves the residual methane pool heavier in $^{13}$C. Our measured methane stable isotopic ratios ($\delta^{13}$C and $\delta^{2}$H) generally clustered at values suggesting a mix of acetoclastic (reduction of organic carbon) and hydrogenotrophic (reduction of carbon dioxide) methanogenesis. HA10 samples deviated from the clustering, suggesting either high levels of microbial oxidation or a thermogenic methane contribution, likely from outgassing of the underlying Kishenehn shale formation. We therefore also measured concentrations of higher chain hydrocarbons—ethane and propane—in these three wells (Fig. 2b). The high ratios of methane concentrations to concentrations of these higher chain hydrocarbons suggested that the aquifer did not contain a thermogenic methane subsidy[15,18,19]. However, none of the samples from which we were able to measure ethane and propane concentrations coincided with heavy carbon isotopic ratios. Therefore, the cause of the heavy isotopic ratios of methane was still unresolved and the possibility of a thermogenic methane subsidy remained valid.

Of the three wells with methane present, only HA10 and HA12 had high enough concentrations to determine radiocarbon ages. The methane in well HA10 was consistently older than that of

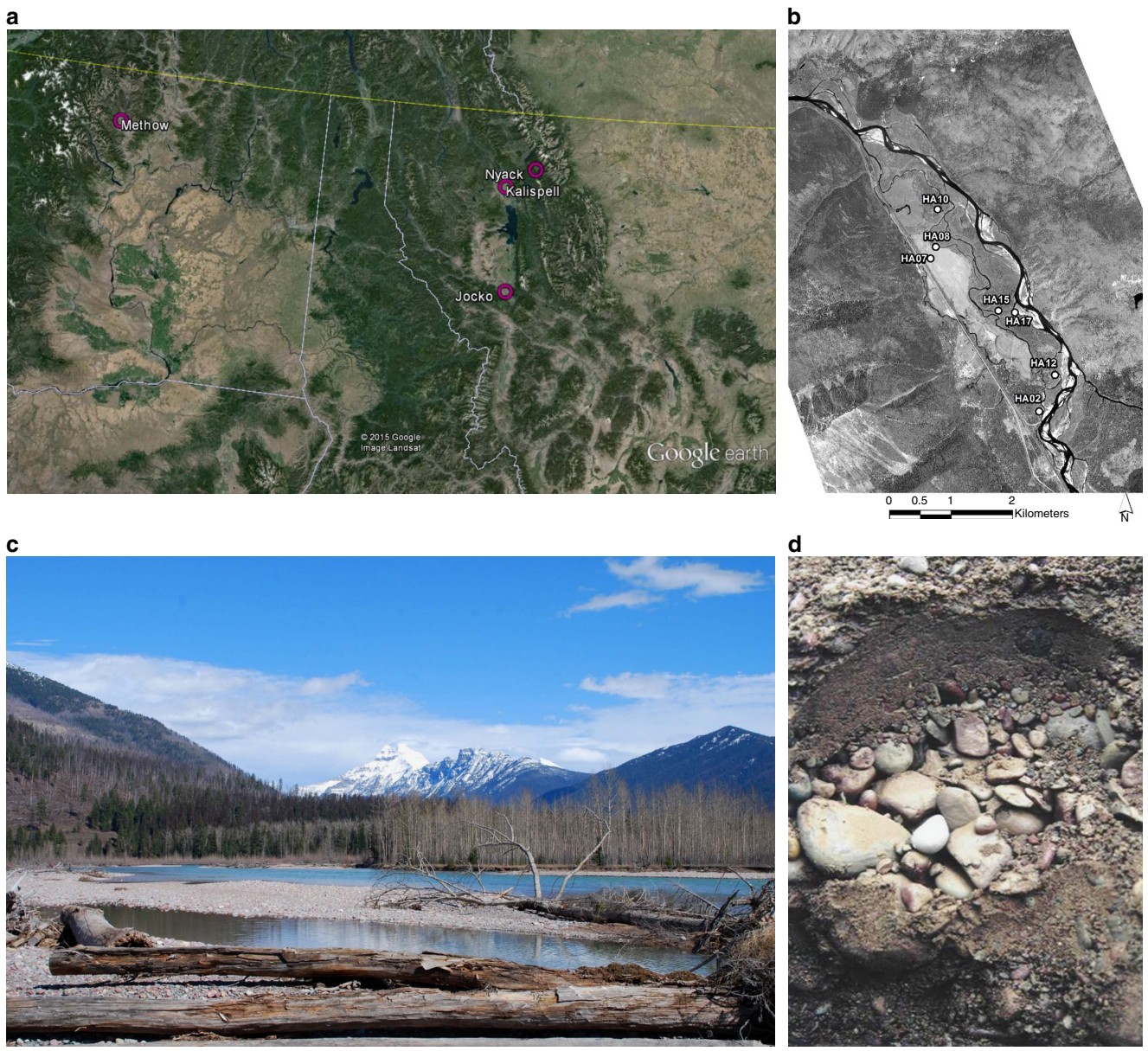

**Figure 1 | Floodplain locations and characteristics.** (**a**). The four floodplains studied are overlaid on Google Earth Imagery. The main research site was the Nyack Floodplain. (**b**). Aerial imagery of the Nyack Floodplain shows the locations of the 7 wells studied (see Supplementary Table 1). (**c**). A view of the Nyack floodplain, near well HA02, shows the pristine nature, landscape complexity, and spatial heterogeneity typical of Nyack. (**d**). A cross-section of the Nyack bed-sediments highlights the heterogeneity of the matrix: sorted cobbles allow extreme hydraulic conductivity, while the fine sediment presents the opportunity to retain organic matter and develop localized hypoxia or anoxia[6].

HA12, and all methane samples that we aged corresponded with methanogenic $^{13}$C signatures and a lack of measurable higher level hydrocarbons. Methane in HA12 ranged from $335 \pm 15$ years BP to $1970 \pm 20$ years BP, and methane in HA10 ranged from $2350 \pm 15$ years BP to $6910 \pm 140$ years BP (Supplementary Table 2). Because radiocarbon ages of dissolved methane samples are the average ages of all sources of methane present, the highly aged methane from HA10 could have included a substantial proportion of ancient methane that is radiocarbon dead; radiocarbon-dead methane could have come from off-gassed methane from the Tertiary age shale underlying the floodplain. For example, the most aged HA10 sample of 6,900 years BP (11/24/2014) could have included up to 58% radiocarbon-dead methane with 42% modern methane: if we assumed that thermogenic methane had a $\delta^{13}$C value of $-50\%$ (refs 13,14) and that microbial methane had a

$\delta^{13}$C value of $-100\%$ (see methods), then the same sample that had a measured $\delta^{13}$C value of $-70.6\%$ could have included a maximum of 59% thermogenic methane. The closeness of these estimates suggested that this sample in particular could have had a substantial thermogenic contribution, though we did not have measured ethane and propane concentrations from the same day to verify or refute this possibility. We concluded that measurable dissolved methane in the aquifer was mainly produced via microbial methanogenesis of modern and ancient organic matter, but a subsidy from thermogenic methane was likely, at least in the HA10 well.

**Methane-derived carbon in Nyack aquifer stonefly biomass.** Five species of amphibitic stoneflies were very abundant in our

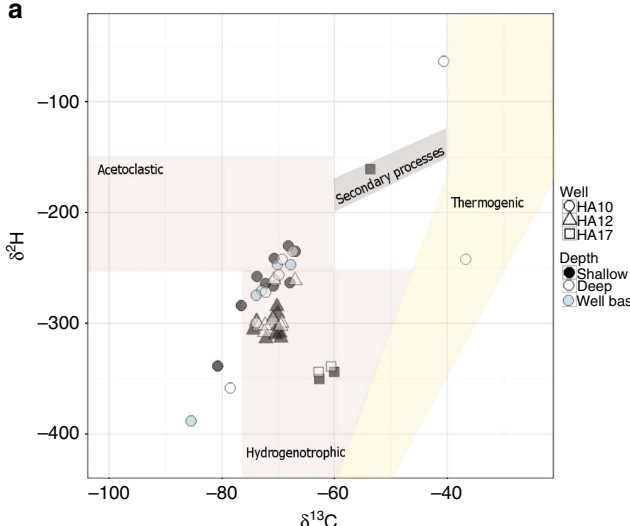

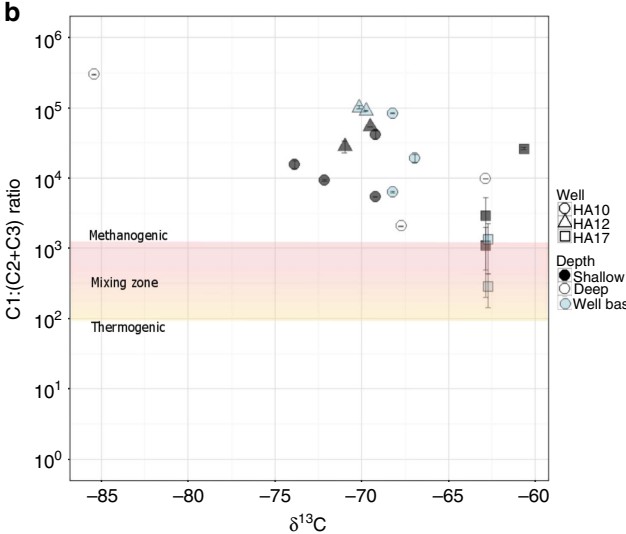

**Figure 2 | Determination of methane origin using stable isotopes and methane to ethane and propane ratios.** (**a**). The Schoell plot (22) of deuterium isotopic signatures vs. carbon isotopic signatures in individual samples. Symbols represent well; colours represent depth. Most samples cluster at a methanogenic origin, while others at HA10 deep and HA17 shallow suggest a thermogenic contribution and/or microbial oxidation. However, samples from the same day at other depths still cluster with methanogenesis. (**b**). A Bernard plot (24) displays the ratio of methane concentration to summed concentrations of higher chain hydrocarbons (ethane and propane) versus the $\delta^{13}C$ of methane.

samples from the Nyack wells: *Paraperla frontalis, Isocapnia grandis, I. crinita, I. integra* and *Kathroperla perdita*[3]. These stoneflies were by far the largest-bodied organisms living in the aquifers. Mature nymphs of *P. frontalis* and *K. perdita* are 2.5–3 cm long, and have mouth parts typically associated with carnivory (elongate mandibles with sharp apical teeth). The *Isocapnia* spp are smaller, 1–2 cm long and have mouthparts characteristic of grazers (short mandibles with short and stout apical teeth)[20]. However, guts of all species contained amorphous, particulate organic-matter (POM), especially in early instars. We concluded that in these dark, organic-carbon-limited aquifers, these large consumers eat whatever organic matter they encounter. Stonefly samples from each species had a wide range of variation in $\delta^{13}C$ (Table 1). This variation indicated that the

stoneflies were consuming methane-derived carbon, rather than deriving low $\delta^{13}C$ values from a symbiosis with methanogenic microbes[21]. Consumption of methane-derived carbon would occur by stoneflies directly or indirectly (via additional trophic linkages) consuming methane oxidizing bacteria (MOB), likely entrained in the amorphous POM. Stonefly biomass $\delta^{13}C$ values were significantly different between species and between dates of collection (ANOVA analysis F test $P < 0.05$) but we observed that the abundance of each species was not consistent across wells. To isolate differences between wells of collection, we pooled both species and dates of collection by well for all subsequent analyses.

We used standard linear two-source mixing models to determine the methane contribution to stonefly biomass in the aquifers[22]. We accounted for methane carbon isotope fractionation by MOB by implementing the most conservative possible estimate of the MOB $\delta^{13}C$ signature as our lower boundary, and the average of our methane $\delta^{13}C$ signatures as an upper boundary, terming these our 'conservative' and 'average' estimates of methane-derived carbon in biomass, respectively (see Methods). We found that stonefly biomass from all wells, including those wells with no measurable methane, included methane- derived carbon (Supplementary Fig. 4). Using a stratified average of both the conservative and average estimates of methane-derived carbon in biomass at each well on the floodplain, we determined that 37.3–66.5% of Nyack aquifer stonefly biomass carbon was methane-derived. Our results therefore showed that the amphibitic stoneflies, the top consumers in the specious aquifer food web at Nyack, were substantially dependent on methane-derived carbon.

**Biomass subsidized by ancient carbon..** Wells HA10 and HA12 were the only wells from which we were able to date the dissolved methane because methane concentrations were low to undetectable in the other wells. Thus only in these two wells were we able to compare methane and stonefly biomass ages. To measure a biomass age most representative of river-supplied carbon, we additionally dated stonefly biomass from the well with the shortest flow path and the lowest overall stonefly methane-derived carbon in biomass: well HA02 (Supplementary Table 1, Supplementary Fig. 5). Biomass radiocarbon ages of individual stoneflies from these three wells were strongly correlated with calculated levels of methane-derived carbon in biomass (log (Age + 1,000) regressed against the average estimate of methane-derived carbon in biomass per individual; $R^2 = 0.56$, F test $P = 2.328 \ 10^{-10}$, $n = 52$) (Fig. 3). This indicated that: (a) a broad range of methane ages was present in the aquifer, (b) the carbon derived from sources other than methane was modern and (c) stoneflies assimilated methane carbon at least 6,900 years BP old.

We used the measured radiocarbon and $\delta^{13}C$ values of stonefly biomass, methane, and organic matter to parameterize a Bayesian mixing model[23] to estimate the contribution of aged or ancient methane to stonefly biomass in all wells. We estimated the distribution of radiocarbon values for organic matter (or all non-methane carbon sources) by weighting stonefly biomass ages by the per cent non-methane contributions calculated from a two-source mixing model of $^{13}C$ signatures. We then created four scenarios considering two possibilities that represented opposite ends of ranges for each methane $\delta^{13}C$ values and the oldest possible methanogenic methane contribution (Supplementary Table 3). Regardless of scenario, the $\delta^{13}C$ values and radiocarbon ages of the stoneflies were significantly different among the three wells (Fig. 4), suggesting that stoneflies were in fact dependent on local food resources that varied spatially. Where river-supplied

**Table 1 | Average and conservative estimates of methane-derived carbon in biomass across floodplains.**

| Floodplain | n Wells sampled | n Insects sampled | Conservative estimate of methane-derived carbon in biomass (%) | Average estimate of methane-derived carbon in biomass (%) | Mean $\delta^{13}$C (%) |
|---|---|---|---|---|---|
| Nyack | 7 | 528 | 37.3 ± 0.1 | 66.5 ± 0.1 | − 55.1 ± 0.1 |
| Kalispell | 6 | 31 | 12.9 ± 0.4 | 23.0 ± 1.2 | − 37.3 ± 0.2 |
| Methow | 4 | 145 | 8.5 ± 0.5 | 15.1 ± 0.2 | − 34.1 ± 0.1 |
| Jocko | 3 | 14 | 20.5 ± 0.7 | 36.5 ± 2.4 | − 42.8 ± 0.4 |

| Species | n Insects sampled | Conservative estimate of methane-derived carbon in biomass (%) | Average estimate of methane-derived carbon in biomass (%) | Mean $\delta^{13}$C (%) |
|---|---|---|---|---|
| l. crinita | 23 | 38.8 ± 3.7 | 69.2 ± 6.7 | − 56.2 ± 2.7 |
| l. grandis | 128 | 33.8 ± 2.2 | 60.2 ± 3.8 | − 52.5 ± 1.6 |
| l. integra | 3 | 14.7 ± 0.9 | 26.2 ± 1.6 | − 38.6 ± 0.7 |
| P. frontalis | 423 | 33.1 ± 1.0 | 59.1 ± 1.7 | − 52.0 ± 0.7 |
| K. perdita | 95 | 18.3 ± 1.6 | 32.5 ± 2.8 | − 41.2 ± 1.1 |
| Isocapnia spp. | 34 | 31.3 ± 3.6 | 55.9 ± 6.4 | − 50.7 ± 2.6 |

Average and conservative estimates of methane-derived carbon in biomass across floodplains were computed as stratified means ± s.e. Sample sizes varied because (a) available sampling wells varied among sites, and (b) stonefly abundance varied among wells within sites. *Isocapnia spp.* includes larvae of *I. grandis* and *I. crinita,* which could not be taxonomically segregated in the larval stage, but were very abundant as easily recognizable teneral adults in both wells.

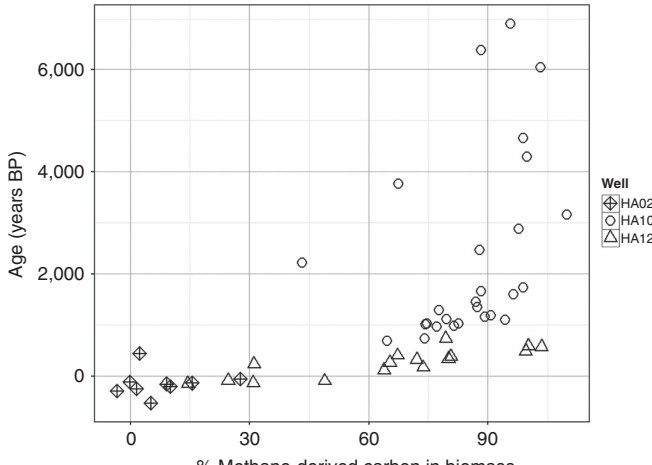

**Figure 3 | Radiocarbon age versus methane-derived carbon contribution to stonefly biomass in three of the wells.** Radiocarbon ages of stonefly tissue (each point is one individual) were strongly correlated with calculated levels of methane-derived carbon in biomass ($R^2 = 0.56$, $P = <0.0001$, $n = 52$), suggesting that (a) a broad range of methane ages was present in the aquifer, as shown by the variation in radiocarbon age even at levels of methane-derived carbon >60%; (b) non-methane-derived carbon was modern because low levels of methane-derived carbon in biomass approaching 0% corresponded with younger radiocarbon ages; and (c) the maximum methane age could be much older than 6900 years, because all stonefly tissue measured was a mixture of various organic carbon sources.

carbon was in lower availability (that is, in wells HA10 and HA12 with longer flow paths, Supplementary Table 1) stoneflies were dependent on methane-derived carbon. Only individuals collected from HA10 relied on aged to fossil methane, supporting the possibility that an ancient shale methane subsidy might indeed exist at this location. In any case, it was clear that the top consumers in the food webs of the Nyack aquifer used a methane subsidy from ancient carbon.

**Methane dependency is widespread.** We collected dissolved methane concentrations and stonefly samples from wells across the three other river floodplains to understand if a methane subsidy to

the groundwater food web was a widespread phenomenon. *P. frontalis, K. perdita* and *I. grandis* were present at all floodplains. We used the same standard two-source mixing model on $\delta^{13}$C values used on Nyack to estimate ranges of methane-derived carbon in stonefly biomass. We parameterized the model using the source estimates for organic matter and methane calculated at Nyack. We found higher estimates of methane-derived carbon in biomass at Nyack than at any other floodplain, but overall methane contributions were high across the other floodplains as well, ranging from 8.5 to 36.5% (Fig. 5, Table 1). This was surprising given that, of fifteen wells analysed across all other floodplains, only three (two at Methow, one on the Jocko) had measurable dissolved methane concentrations (Supplementary Fig. 5). This was similar to the case we found at Nyack, where methane-derived carbon in biomass existed at all wells regardless of methane concentrations.

**Discussion**. The data showed that the Nyack food web was heavily subsidized by methane with various carbon ages (from modern to millennial aged or fossil), most of which was methanogenically produced. Although we could not verify a thermogenic methane contribution through presence of ethane and propane concentrations, the documented existence of carboniferous shale at Nyack[8], presence of highly aged carbon, and presence of heavy methane $\delta^{13}$C added credence to the possibility of a thermogenic contribution to the aquifer food web. Because methanogenesis occurs mainly in anoxic environments and MOB flourish in opposing gradients of methane and oxygen[24], it is likely that stoneflies were directly or indirectly consuming resources produced at these interfaces. In fact, the heaviness of biofilm and organic matter $\delta^{13}$C signatures relative to stonefly biomass signatures suggested that stoneflies preferentially consume methane-derived carbon. This could explain their abundance in such a carbon-limited system. Furthermore, because the amphibitic stoneflies emerge from the river as flying or crawling adults, they are exporters of labile organic carbon from the aquifer to the floodplain surface, as well as top consumers in a food web that clearly sequesters methane, a powerful greenhouse gas.

All of the floodplains which we studied had substantial site-wide methane subsidies to top consumers, and Nyack additionally had a millennial-aged to fossil methane subsidy. There are multiple possibilities for the origin of the millennial-aged to ancient methane at Nyack: if methanogenic, it could have been produced from buried organic matter

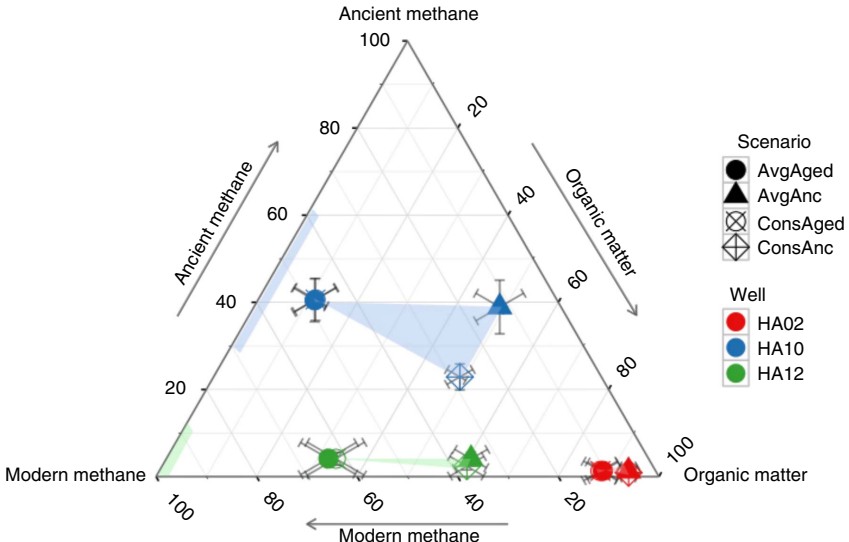

**Figure 4 | Bayesian modelling outcomes showing carbon contributions of methane and organic matter to stonefly biomass.** Per cent contributions from each modern methane, ancient methane, and organic matter were plotted for each well (colours) for each of the four mixing-model scenarios explained in text and Supplementary Table 4 (symbols). The shaded areas represent the full range of possibilities for source contributions considering the four scenarios. Error bars represent s.d.'s of each estimate. The shaded lines on the methane axis (left) represent the potential mixtures of modern and ancient methane in each well from which we could measure methane ages (HA10 and HA12). Wells HA10 and HA12 were the only two wells on the floodplain with high methane concentrations, and well HA02 was closest to the river with the shortest flow path and lowest levels of methane-derived carbon contribution to stonefly biomass across all samples.

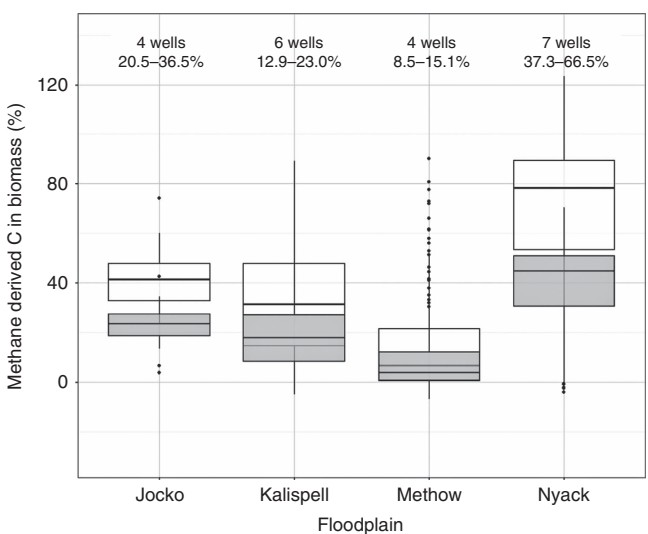

**Figure 5 | Methane-derived carbon contributions to stonefly biomass across floodplains.** Boxplots of methane-derived carbon contribution to stonefly biomass for each of the floodplains studied using both the average and conservative estimation techniques (see Supplementary Table 1 for the two estimates). These estimates each assumed an extreme end of a range of potential source $\delta^{13}C$ values for methane (see text). The values displayed above each bar are the average methane-derived carbon in biomass values (stratified by well) for each floodplain. Boxes represent the interquartile range with median and whiskers extend to 1.5 times the inter-quartile range for each set of floodplain estimates.

deposited since the last glaciation[25]; it also could have come from outgassing of thermogenic methane.

This is the first report of a methane-derived carbon contribution to top consumer species across multiple river ecosystems, and the first report of an ancient methane-derived carbon contribution to any freshwater consumers. While methane

cycling and aged carbon have each been studied in rivers[26–32], the few published studies that document a river food web methane subsidy are site-specific and do not report that the methane-derived carbon was millenial-aged or ancient. For example, Caraco et al. documented an ancient carbon subsidy to zooplankton in the Hudson River Estuary[28], Kohzu et al. showed that some macroinvertebrate production was fuelled by biogenic methane produced from detritus in backwater pools[29], and Trimmer et al. showed that caddis fly species derived up to 30% of their biomass from methanogenic methane in the River Lambourn[31]. Additionally, the aquifers that we studied are dark. Therefore, the $\delta^{13}C$ depletion in stonefly biomass could not have occurred from fractionation of isotopically light $CO_2$ during photosynthesis, as has been found in previous studies[33].

River floodplains are among the most threatened ecosystems in the world owing to dams, revetments and gravel mining, among many other perturbations[34]. The surprising details of groundwater ecology described herein provide a broader basis for river protection and conservation. The amphibitic stoneflies are prolific denizens in coupled river-aquifer ecosystems characterized by unperturbed spatial and biogeochemical complexity that produces exploitable food web carbon while at the same time providing natural filtration processes that maintain water quality for downstream reaches.

## Methods

**Sample collection and processing.** The Nyack floodplain was equipped with seven 3-inch PVC wells with 2 mm slot openings down the length of the pipe. The wells were drilled 8–10 m using a hollow auger drilling rig (Supplementary Table 1)–for aquifer characteristics measured at each well. Wells HA02, HA07, HA08, HA10, HA12 and HA15 (Fig. 1) were equipped with sensors and data loggers that recorded dissolved oxygen, temperature, depth and specific conductance on an hourly basis. We used a peristaltic pump equipped with PTFE (Teflon) tubing to draw water from two to three depths at each well—1 and 4 m below the baseflow water table at all wells, with an additional depth of 0.5 m above bedrock (well base) at well HA10. The Kalispell floodplain had seven existing wells drilled similarly (slotted but not screened, and drilled to maximum possible depth). The Jocko had three wells, and the Methow four. We sampled these wells also at 1 and 4 m below the base flow water table.

We sampled methane concentrations approximately every three weeks at all wells on Nyack and in Kalispell for two years, from August 2013 to August 2015.

We sampled four times during 2014–2015, once in July 2014 and four times from March to September 2015 on the Jocko and on the Methow. We used a modified active-sampling method: we pumped sample water into a BOD bottle, allowing it to overflow for one to two minutes before withdrawing 1–7 ml using a 22-gauge needle attached to a two-way stopper and 10 ml syringe. In the lab, we had capped 9.83 ml glass scintillation vials with PTFE-lined grey butyl stoppers and crimp seals, then flushed them three times with ultra-high-purity $N_2$. We simultaneously injected field samples to the sample vials while allowing excess $N_2$ to drain into a second syringe. We poisoned the samples to 0.5% $ZnCl_2$ and stored them upside down at 4 °C until analysis within one week of collection. We analysed samples on a greenhouse gas chromatograph (SRI Instruments model 8610C) equipped with a flame ionization detector and SRI PeakSimple Software. We calculated headspace methane concentrations using a three-point calibration with Scotty gas standards (Air Liquide America). We then used Henry's Law to calculate dissolved methane before headspace equilibration using the solubility constant documented by Yamamoto et al. (1976)[35]. Error averaged 0.08 $\mu$mol l$^{-1}$ initial aqueous concentration and our detection limit was 0.11 $\mu$mol l$^{-1}$.

We collected samples for measurement of ethane and propane concentrations beginning in Spring 2015 using similar methods as for methane collection. The discrepancy in timing arose because only by Spring 2015 did our stable isotope results reflect a potential thermogenic contribution. We used similar methods as for collection of samples for measurement of dissolved methane concentrations, but instead injected 30-ml samples to 38.25 ml glass serum vials capped with thick black butyl stoppers. We did this to minimize our detection limit of ethane and propane by maximizing the equilibrium headspace concentrations of ethane and propane, which we expected to be present at low concentrations if at all. We used Henry's Law and solubility constants documented by Hine and Mookerjee[36] to calculate the dissolved concentrations of ethane and propane in waters.

We collected samples for analysis of dissolved methane stable isotope composition from wells HA10, HA12 and HA17 using acid-washed Teflon tubing and the same active sampling methodology used for collecting concentration measurements. However, we instead injected samples into evacuated Exetainers vials (Labco Limited), then similarly poisoned them to 0.5% $ZnCl_2$. We sent samples to the University of California at Davis Stable Isotope Facility for $^2$H and $^{13}$C analyses, where they were analysed on a Thermo Scientific Delta V Plus isotope ratio mass spectrometer (IRMS, Thermo Scientific, Bremen, DE) according to the methods of Yarnes et al.[37] Long-term s.e. was 0.2% for $\delta^{13}$C and 2% for $\delta^2$H.

When methane concentrations were at a minimum of 10 $\mu$mol l$^{-1}$, we were able to collect samples for radiocarbon analysis of dissolved methane. We pumped water into an acid-washed and baked 1 l glass microculture bottle, keeping the hose at the bottom of the bottle and timing until it filled. We then inverted the bottle underwater and allowed water to flush through for the duration of time it took for the bottle to fill; this also allowed for the capture of any outgassed components. We capped the bottle in this state with blue butyl and crimp seals, poisoned the sample with 10 ml 50% $ZnCl_2$, and transported it back to FLBS on ice, where we injected a 4 ml UHP $N_2$ headspace and shipped samples to UC-Irvine's Keck Carbon Cycle AMS facility.

Upon arriving at the UC-I Keck lab, ~15% headspace was created in the sample bottle by injecting ultra zero air with a syringe and removing the displaced volume of water in the same time. Samples were shaken for 1 min and allowed to settle for 30 min before extraction. An evacuated 2 l stainless steel canister attached to a needle was used to extract headspace gases from the water bottle. The canister was then filled with 1 atm pressure with ultra zero air, which served as a carrier gas in the latter extraction. On a flow-through vacuum line, the headspace $CH_4$ and $CO_2$ were separated, combusted and purified[38], and graphitized by the sealed tube Zn reduction method[39] then measured for radiocarbon ($^{14}$C) on a compact accelerator mass spectrometer (AMS, National Electrostatics Corp.)[40]. For dry stoneflies, the samples were weighed into prebaked quartz tubes with prebaked CuO, evacuated, sealed then combusted at 900 °C for 2 h. After combustion, sealed tubes were cracked and $CO_2$ was extracted on a vacuum line, graphitized and measured for $^{14}$C using the same method mentioned above. Data presented here are expressed as radiocarbon age (year, BP) and $\Delta^{14}$C (%) as well. Both were normalized to radiocarbon activity of an oxalic acid standard OX1 and isotopic fractionation corrected to $-25$% (ref. 41). For $\Delta^{14}$C, standard OX1 was also decay corrected to 1950. $\Delta^{14}$C (%) = / > 0 can be used to indicate 'modern' carbon (1950 to present) and $\Delta^{14}$C (%) < 0 for 'aged' carbon (pre-1950) and $\Delta^{14}$C (%) = $-1,000$% for 'fossil' or $^{14}$C dead carbon. The $\Delta^{14}$C analytical error was ~2% for modern sample, based on long-term measurements of secondary standards.

$\delta^{13}$C analysis was made on $CO_2$ subsampled from the vacuum line and measured by using a Gas Bench II coupled with a Thermo Scientific Delta Plus XL IRMS. The $\delta^{13}$C analytical error was ± 0.15% based on long-term measurements of secondary standards.

On the same days when we collected methane samples, we collected as many stonefly nymphs as possible via trapping methods. To trap, we suspended nylon ropes in the wells on which the emergent and resident stoneflies could climb. We checked the ropes every two weeks and collected larvae if present. Every six weeks, we additionally pumped the wells using a gas-operated diaphragm pump. Samples were kept at a minimum of three metres from the pump while the pump was running to avoid potential contamination. All output water was transferred through 2.5′ Tigerflex tubing and emptied into a 330 micron Nitex mesh net. We elutriated the retained samples, collecting stoneflies caught in the net and

transferring them to distilled water (DI). We kept the stoneflies at 4 °C for a minimum of 24 hr to clear gut contents, then identified them to species level[20,42], rinsed them in DI water, and transferred them into individual sterile cryovials. We froze samples and stored them at $-80$ °C until preparation for stable isotope analysis. We took the same collection approach for collecting organic matter and biofilm samples for stable isotope analysis, but samples were collected in June to July 2013. We used 64 and 500 $\mu$m Nitex mesh to parse out fine and coarse organic matter, respectively. These samples were also frozen until preparation for stable isotope analysis.

We collected biofilm samples by suspending ashed and autoclaved gravel bags at all sampling depths for ten weeks during July and August 2013. We collected samplers into sterile Whirl-paks and froze them. To remove biofilm and strongly associated particulate matter for stable isotope analysis, we defrosted and filled Whirl-paks with 200 ml ultra-pure distilled water, then sonicated for 40 min. We poured the solution into sterile glass beakers, rinsed the remaining gravel into the beakers, and dried the beakers at 60 °C until water evaporated (usually 3–4 days). We then scraped the samples into silver capsules and acidified them by fumigation with hydrochloric acid[43].

To prepare stoneflies and organic matter for stable isotope analysis, we defrosted them at room temperature, rinsed the stoneflies a second time in DI, and transferred them directly to aluminium foil to dry at 60 °C for at least 48 h. We then milled them into a fine homogenous powder using steel milling capsules and a grinding mill for 20 seconds each. We subsampled 0.8 to 1.2 mg of each stonefly into tin capsules and replicated approximately once per 15 samples (replicate coefficient of variation = 0.2%). Samples were analysed on a PDZ Europa ANCA-GSL elemental analyser interfaced to a PDZ Europa 20–20 isotope ratio mass spectrometer (Sercon Ltd., Cheshire, UK) at the UC Davis Stable Isotope Laboratory. Stable isotope ratios were expressed relative to international standards: V-PDB (Vienna PeeDee Belemnite) for $^{13}$C and air for $^{15}$N.

**Methane source determination.** To calculate potential methane source contributions, we considered three sources: modern methanogenic methane, ancient methanogenic methane, and thermogenic methane (for example, shale). Both classifications of methanogenic methane are biologically produced anaerobically through either acetoclastic methanogenesis or hydrogenotrophic methanogenesis. In the former scenario, methanogenic archaea require an organic carbon source. The second scenario is an autotrophic process in which methanogenic archaea produce methane from carbon dioxide and hydrogen. In both forms of methanogenesis, the methane produced is drastically depleted in $^{13}$C ($-50$ to $-80$%) due to methanogens preferentially assimilating lighter carbon isotope in their metabolism ($^{12}$C) (ref. 14). Methane in freshwater systems can also be released from thermogenic sources such as shale or coal, though this has not previously been documented as an ecological subsidy. In this case, hydrocarbons are produced as a result of abiotic pressure and temperature conditions. Thermogenic methane carbon and hydrogen are both isotopically heavier than the methanogenic methane, and the thermogenic methane is usually accompanied by higher level hydrocarbons such as ethane and propane[13,15]. It is also radiocarbon-dead or greater than 50,000 years in age. Ancient organic matter can also be methanogenically decomposed to produce highly aged methane. All methane can then be consumed by MOB, which fractionate the dissolved methane by preferentially assimilating the lighter carbon isotope, leaving the residual methane enriched in the heavier isotope. In the exponential phase of MOB growth, fractionation in MOB biomass is 30.3% (ref. 19). During normal growth phases, fractionation is 16% (refs 17,19). A graphical summary of source determination using carbon and hydrogen isotopes is overlaid on Supplementary Fig. 3A. We used isotopic signatures in combination with radiocarbon dating, and measurement of ethane and propane concentrations to determine the methane sources. These results are displayed in Supplementary Fig. 3.

Our results suggested that the majority of dissolved methane was derived from a mixture of acetoclastic and hydrogenotrophic methanogenesis. The samples that deviated from this general classification were both taken from well HA10 at the deep sampling depth. Methane oxidation involves wide variation in deuterium fractionation depending on temperature and all thermogenic methane tends to have heavier deuterium isotopes[15–17]. Therefore, these samples could have resulted from high levels of oxidation or a contribution from a thermogenic methane source. This range of possibilities was reinforced by radiocarbon dating, which showed that dissolved methane collected from HA10 was consistently older than dissolved methane from HA12.

We therefore began to collect samples for the measurement of ethane and propane concentrations in May 2015. These samples corresponded with stable isotope signatures and insect and radiocarbon ages. None of the samples from which we measured ethane and propane concentrations were found to be isotopically heavy despite finding insect biomass carbon consistently aged up to 6,900 years BP. None of the samples we measured had ethane or propane concentrations high enough to suggest a thermogenic methane contribution. In general, if the ratio of methane concentration to the summed concentrations of ethane and propane is <100, then the source is thermogenic. If it is >1,000, then the source is methanogenic[44]. Anything in between is considered a mix. No samples had ratios significantly <1,000 (Supplementary Fig. 3B). We therefore concluded that the samples which we measured had no thermogenic methane

contribution, though we HA10 deep might still have a thermogenic methane contribution that is so minimal and unpredictable that we very occasionally have the opportunity to measure it.

**Causes of stonefly biomass $\delta^{13}C$ depletion.** We also measured the age of dissolved $CO_2$, which ranged from $1310 \pm 15$ to $1970 \pm 20$ years BP, suggesting that older methane carbon contributions were from organic matter rather than DIC which would be similar to the dissolved $CO_2$ (Supplementary Table 2). The $\delta^{13}C$ values ranged from $-19.2$ to $-14.8\%$ ($n = 6$), which further indicates this carbon pool is not likely the main contributor for the stonefly biomass. Although the carbon isotope fractionation indicated by the low $\delta^{13}C$ values in these estimates (Table 1) can occur via other pathways such as ammonium oxidation and sulphur oxidation, the resulting $\delta^{13}C$ values would be far heavier than those we observed. Ammonium oxidation produces bulk biomass depleted in $\delta^{13}C$ by 20% relative to $CO_2$, which we measured as $-16.6 \pm 0.7\%$ (ref. 45), and sulphur oxidation produces bulk biomass depleted in $\delta^{13}C$ by 24.6 to 25.1% relative to $CO_2$ (ref. 46).

**Methane contribution to biomass: $\delta^{13}C$ models.** Regardless of methane source, it was necessary to account for the variation in isotopic signatures of methane across the floodplain as we proceeded to calculate methane-derived carbon contributions to stonefly biomass. We assumed that stoneflies consumed MOB as is suggested by the large variation in stonefly biomass $\delta^{13}C$ values even within species (Table 1).

We used a two-source mixing model[22] on stonefly biomass signatures to calculate relative contributions of MOB and organic matter using $\delta^{13}C$ values:

$$\% \text{ methane-derived carbon in biomass} = \frac{\text{Stonefly}\partial^{13}C - \text{OM}\partial^{13}C}{\text{Methane}\partial^{13}C - \text{OM}\partial^{13}C} \times 100$$

To represent any possible contribution of organic matter to stonefly diet, we used 'organic matter' as a surrogate for any component of the stonefly biomass that was not methane-derived carbon. Means and s.e. of $\delta^{13}C$ values for each organic matter classification are displayed in Supplementary Table 4. Coarse particulate organic matter (CPOM) showed depletion relative to other organic matter pools because stonefly detritus was inevitably and visibly incorporated into the CPOM pools we collected via pumping. We used a stratified average of all OM pools, $-27.83 \pm 2.49\%$, which is approximately the literature estimate of photosynthetically fixed terrestrial carbon: $-28\%$ (ref. 22).

To estimate the $\delta^{13}C$ value of MOB, we bracketed estimates using our measured values of methane itself and maximum levels of fractionation by exponential growth of MOB ($\alpha = 30.3\%$)[19]. We preferred to use a Keeling plot[47] to estimate methane signatures at the time of production, but our data showed extensive variation in isotopic signatures even in samples collected at times with high methane concentrations, making such an estimation technique unfeasible (Supplementary Fig. 6). We therefore averaged all samples ($n = 32$) collected at times when methane concentration was $> 1 \mu$mol, yielding $-68.79 \pm 8.52\%$. This was termed our 'Average' estimate of source methane $\delta^{13}C$ and therefore a suitable estimate for the heaviest possible isotopic ratio representative of MOB biomass. We then applied the fractionation factor to this estimate, yielding a most conservative estimate (lightest possible isotopic ratio) of $-100.86\%$ using the equation[16]:

$$\alpha_{\text{Product}}^{\text{Source}} = \frac{1000 + \partial^{13}\text{Source}}{1000 + \partial^{13}\text{Product}}$$

We termed values of methane-derived carbon contributions using this estimate as our 'Conservative' estimate. We presented both sets of data in the results.

Estimates of methane-derived carbon in biomass were normally distributed. We found a significant effect of species and date of collection on methane-derived carbon in biomass using simple linear regression models and ANOVA analysis (main text). However, both of these variables were strongly confounded with well of collection, as stonefly life history and well conditions inevitably determined the environment which they inhabited at the time of sample collection and thereby influenced the quantities of each measured. In regards to date, we collected from time points over four seasons and during 1–2 years at all sites to avoid bias from sampling time. In order to compare overall levels of methane-derived carbon contributions to stonefly biomass across and within floodplains, therefore, we only considered wells as strata and pooled species at all times of collection. Please see raw data files for dates of sample collection at each well.

**Methane contribution to biomass: $\delta^{13}C$ and $\Delta^{14}C$ models.** We submitted 52 stoneflies from wells HA02, HA10 and HA12 for combined stable isotope analysis and radiocarbon dating at the WM Keck facility at UC Irvine (see Methods above). We then were able to use both $\delta^{13}C$ and $\Delta^{14}C$ values for implementing a Bayesian framework stable isotope mixing model to infer contributions of various potential methane pools to stonefly biomass. This model considered aged methane, ancient methane, modern methane, and modern organic matter as potential sources, using scenarios of both average and conservative MOB $\delta^{13}C$ values.

We inferred the source values for organic matter by taking a weighted average of $\Delta^{14}C$ values across the 52 stoneflies. Our weights (OM dependence) were calculated as 1- (methane-derived carbon contribution obtained via equation 1). Because we could calculate methane-derived carbon contribution using either the Average (Avg) or Conservative (Cons) approaches, we had two estimates for OM

$\Delta^{14}C$: Avg: $-13.7 \pm 32.2\%$ and Cons: $-65.6 \pm 75.9\%$. We used these in the Avg and Cons scenario types (Supplementary Table 3).

For each of the Avg and Cons scenarios, we also had two estimates for maximum methane age measured using $\Delta^{14}C$. The radiocarbon ages that we measured in methane, ranging from 335 to 6900 years BP, were by definition an average of the various carbon ages present in that methane sample. Each sample was therefore a mixture of methane source ages. We therefore created one methane source to represent modern methane, taken as the Avg radiocarbon age of OM, and a second methane source as either aged or ancient methane. Aged methane was given the $\Delta^{14}C$ value of the oldest measured methane ($-580 \pm 7.2\%$) and ancient methane was considered to be radiocarbon dead, or $>50,000$ years in age ($-1,000\%$). This contributed another dimension to the scenarios needed: Aged and Anc (ancient) methane. Again, the pathway for incorporation of either methane type to stonefly biomass would be via MOB and we therefore needed to consider the possibilities of minimum and maximum fractionation (Avg and Cons). The four scenarios and their associated source values and standard deviations are displayed in Supplementary Table 3.

We implemented the mixing model in the R platform[48] using the SIAR package[23]. The SIAR package allows for the input of source mean stable isotope signatures and their standard deviations. It also requires the input of trophic enrichment factors and their standard deviation, which we took as widely used literature averages[22]. Individual stoneflies were grouped by well. The SIAR package uses a Monte Carlo Markov Chain simulation to calculate a distribution of possible contributions of each source to each group. We ran the model for 10,000 iterations with a burn-in of 1,000 runs for each scenario. We then compiled the run results and calculate mean and standard deviations of each source contribution to biomass in each well analysed (HA10, HA12 and HA02). Results for the four scenarios are all displayed in Fig. 4 (main paper).

**Data availability.** Data referenced in this study are tabulated in Supplementary Tables, and deposited at figshare, DOI 10.6084 (10.6084/m9.figshare.3519782), or available on request from the corresponding author (AGD).

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

## Acknowledgements

We thank Adam Baumann, Bonnie Ellis, Diane Whited, Hannah Coe, Jon Graham, Ashley Helton, Shawn Devlin, Geoffrey Poole, Steve Whalen, Chad Reynolds, Brian Reid, Meredith Wright, Anne Hershey, FLBS faculty and staff, volunteers and the Dalimata family for help throughout the project. We also thank anonymous reviewers who helped to improve the manuscript. Funding was provided by the Jessie M. Bierman professorship and philanthropic donations.

## Author contributions

A.G.D. and J.A.S. designed the study and conducted data collection, data analysis and writing. X.X. contributed to sample analysis and writing.

## Additional information

**Competing financial interests:** The authors declare no competing financial interests.

