## [Peer Review File · Nature Communications]

Reviewers' comments:

Reviewer #1 (Remarks to the Author):

Comments on DelVecchia et al. Nature Communications.

This is a wonderful study and very clearly written and presented. The authors show that top predators (stoneflies) in streams in Montana and Washington are made of very old, highly ^{14}C depleted carbon that comes from old methane. The authors use a combination of radioisotope, ^{13}C and deuterium (^2H) analyses to support this conclusion. While ecosystems dependent on old methane have been demonstrated in the ocean in so-called "methane seeps" this is the first report of support of a freshwater system by aged methane.

line 394- The authors state they measured the age of dissolved CO_2 . I think they mean that they measured the age of dissolved inorganic carbon which includes CO_2 , HCO_3^- and CO_3^{2-} . It is possible using the equations in Zhang to derive the ^{14}C contents of the individual C species from the measurement of ^{14}C in DIC but it is not clear if this is what the authors did. Please clarify. Also, if they did mean ^{14}C of DIC, they would need to do these calculations because the ^{14}C of the CO_2 moiety is more depleted (e.g. older) than for the DIC in toto.

Figures. There are many more figures than there are figure legends and so I am confused here about which figures the authors intend to be in the paper. This is messy and makes the paper difficult to fully evaluate.

Reviewer #2 (Remarks to the Author):

This is a very interesting study tackling two timely issues in freshwater science: methane as a carbon source and the age of carbon entering aquatic food webs. You present quite compelling data to demonstrate that larval biomass of a number of stonefly species inhabiting flood plain aquifers is made up partially of methane-derived carbon (MDC) and that part of that carbon is ancient. These are important findings that will be of interest to others in the field and potentially to ecologists more widely. However, although the findings of the study certainly extend current knowledge of both the two issues, both of which have been attracting publications in recent years, I think you generally rather overstate the novelty of your work. The most novel aspect is how ancient carbon may be transferred via biogenic methane and MOB to consumers in this floodplain system. Unfortunately the data cannot resolve whether the methane itself is "ancient" or if it is "modern" methane produced biogenically from "ancient" organic matter (no data are provided on DOM from which any methane would likely be produced given that you reasonably convincingly discount the alternative of methane production from ancient CO_2) and this does weaken the impact and significance of the study. Moreover, the pathway via MOB is inferred and, while I do not really doubt it, you have not actually demonstrated the presence of MOB in the diets of the invertebrates or even in the system at all, which again is a weakness. Overall the approach and methodology are valid, but some data are lacking (isotope values for DOM and other invertebrates in the system; direct evidence for presence of MOB) that would have been very helpful and could have provided greater insight into the mechanisms/pathways involved. So far as I can judge the statistical approaches are sound, and the use of the Bayesian models allows uncertainties to be incorporated. The conclusions are generally valid and robust (aside from the weaknesses already referred to), but the language needs to be tightened to make the inferences/conclusions more precise. The paper is reasonably well constructed, but was not always easy to follow. One problem is the frequent references in the text to tables and figures that are included as Extended Data; I think several of these items deserve to be within the

main body of the paper and immediately available to readers rather than presented as supplementary information. Conversely, Figures 1 and 2 were of negligible value in understanding the text and could be deleted or at least moved to the Extended Data category. For a reader like me not familiar with these floodplain aquifer systems and their hydrology, a schematic diagram might be very helpful to illustrate the hydrologic pathways, the habitat/location of the animals and the potential pathways for methane and organic matter input into the system.

Overall, although I enjoyed reading the paper and found the findings of considerable interest, I am afraid there are too many shortcomings and gaps in the narrative for me to recommend publication in Nature Communications. Some of these could certainly be addressed in a revision, but others are more intractable and at the very least would require additional data. I provide some more detailed specific comments below, listed by line number (L).

L1 In the title and throughout the text I am not happy with your use of the term "ancient methane". To me this implies that the methane itself is ancient, and this is just not proven by the 14C aging. The methane could have been produced recently from ancient organic matter but still register the same 14C age. I think the way you present the results in this context is sloppy and you need to tighten the expressions to avoid misleading inferences.

L14 I think you should say "much of the biomass carbon...is methane-derived." It is not all methane-derived, and your estimates range from as high as 66% to as low as 15% for those animals you have analysed.

L15-17 Here you cannot say anything about the age of the methane - it is the age of the carbon in the methane. Also I think you could say "apparently mostly derived from biogenic sources although some thermogenic contribution cannot be excluded" to better reflect your data.

L18 What does "the most extensive methane-dependent freshwater ecosystem ever described" mean? Seems like hyperbole!

L24 This opening sentence is a bit odd, as it is not normal practice to mention journal titles when citing literature. Seems like journal name dropping! Also, as the papers are now several decades old it is enough to say they "revolutionized" rather than "have revolutionized".

L43 You describe the stoneflies as top consumers but in ED Fig 1 they are categorised as grazers/omnivores, so this perhaps warrants some explanation as to why there are no predatory top consumers in this ecosystem.

L43 The legend to ED Fig 1 states that 17 Plecopteran species occur in the Nyack floodplain, but the figure only appears to indicate 2 species. The legend then states that 5 species commonly occurred in the well samples and these are listed in Table 1. Why do these numbers not tally?

L54 You mention a paucity of labile OC, but what about total OC? Methane can be produced biogenically from DOC that would not be considered labile.

L133 "but the Isocapnia spp are thought to be grazers" - thought by whom?

L135 Yes, it indicates a likely role for MOB but does not directly prove it, and cannot resolve whether the stoneflies are using MOB directly or indirectly by an additional food web link.

L197 To reiterate, it is the carbon in the methane that represents different ages, not necessarily the

methane itself.

L202 on. This inference may very well be correct, but it is really entirely speculative. You have not directly demonstrated the presence of MOB in the system, or even demonstrated that methane oxidation is occurring. If stoneflies directly and preferentially graze the MOB how do they do this? Your $\delta^{13}\text{C}$ values for biofilm indicate that if MOB are present in the biofilm they must be a tiny proportion (do you have any microscopic or DNA evidence regarding the composition of the biofilm?), so it is difficult to see how stoneflies could preferentially graze them, especially those species with mouthparts "typically associated with carnivory" (L132). So the MOB are presumably located elsewhere, but then how might the stoneflies access them? Or is it possible that the stoneflies use them indirectly by consuming an intermediate trophic link? Here it would have been really useful to have $\delta^{13}\text{C}$ data for other taxa from those shown in ED Fig 1 to see if any of those also show low $\delta^{13}\text{C}$ values indicative of MOB incorporation. It would also have been very valuable to have some independent corroboration of MOB presence and use (directly or indirectly) by the stoneflies; for example, fatty acid analysis of stonefly tissue could reveal the presence of fatty acids that serve as specific biomarkers for MOB and this would unambiguously confirm both the presence of MOB in the system and a role for them in stonefly nutrition.

L210 Surely methane at the very low methane concentrations recorded does not represent a "potential water contaminant".

L211-215 I really think you overstate the case here in several regards.

L215-225 I think you underplay previous work. The study by Caraco et al showed a role of ancient OM in the Hudson River, and ancient OM could contribute to your system via methanogenesis and then MOB. It is not true that all previous studies have been site-specific. Reference 26 (Shelley et al) assessed methanotrophy in relation to photosynthetic PP across a wide range of rivers in southern England; it is true that that paper did not measure methane contribution to consumers, but the implications are clear. It is not fair to imply that lake studies of methane contributions only relate to north-temperate lakes; there have been reports from Arctic and tropical lakes, and Jones et al (2008, Ecology) presented data from almost 100 lakes with a global distribution. Besides the "few midge species" that have attracted most attention, there are reports of several other benthic taxa that appear to contain some methane-derived carbon based on their low $\delta^{13}\text{C}$, and also several reports concerning crustacean zooplankton with $\delta^{13}\text{C}$ values lower than those you report for stoneflies, and even reports showing transfer of MDC up to fish.

L 231 I don't think you can say the stoneflies are "methane dependent". They may use MDC, in some instances to quite a large extent, but methane-dependence surely implies that they could not manage without methane, and I do not think that is really the case.

L329 It is unfortunate that your $\delta^{13}\text{C}$ values for biofilm come from these 10 week incubations of gravel bags. I wonder how well these represent the taxonomic and isotopic composition of the long-term in situ biofilm. Note also that the isotope values of these slowly maturing large stoneflies will reflect diets over quite a long span of preceding time whereas your incubated biofilms have "current" isotopic signatures.

L356, Strictly, MOB do not "prefer" the lighter isotope; rather the lighter isotope is preferentially utilised in their metabolism.

L394-398 Some other studies have been complicated by the possibility that methane carbon could enter food webs by oxidation to heavier but still very light CO_2 which is then incorporated into algae

by photosynthesis and then passed to consumers; so not by direct consumption of MOB. In fact van Duinen et al (2013, Freshwater Science) argued that this was the predominant pathway by which MDC reached invertebrates in Estonian bog pools. In your dark hyporheic system this is presumably not a possible pathway, and the data you present for CO₂ nicely confirm this. I think you should make this point more strongly in the paper as not all potential readers will necessarily appreciate it otherwise.

L422 Specify "photosynthetically fixed TERRESTRIAL carbon" - aquatic photosynthesis can produce a much wider range of d¹³C values for organic matter.

32125 Bio Station Lane

Polson, Montana, U.S.A. 59860-6815

Phone (406) 982-3301

Fax (406) 982-3201

<http://flbs.umt.edu>

<http://flbs.umt.edu/webcams>

flbs@flbs.umt.edu

We appreciated the constructive criticisms of the reviewers and have addressed their concerns below.

Reviewer Comments:

Reviewer #1 (Remarks to the Author):

Comments on DelVecchia et al. Nature Communications.

This is a wonderful study and very clearly written and presented. The authors show that top predators (stoneflies) in streams in Montana and Washington are made of very old, highly ^{14}C depleted carbon that comes from old methane. The authors use a combination of radioisotope, ^{13}C and deuterium (^2H) analyses to support this conclusion. While ecosystems dependent on old methane have been demonstrated in the ocean in so-called "methane seeps" this is the first report of support of a freshwater system by aged methane.

line 394- The authors state they measured the age of dissolved CO_2 . I think they mean that they measured the age of dissolved inorganic carbon which includes CO_2 , HCO_3^- and CO_3^{2-} . It is possible using the equations in Zhang to derive the ^{14}C contents of the individual C species from the measurement of ^{14}C in DIC but it is not clear if this is what the authors did. Please clarify. Also, if they did mean ^{14}C of DIC, they would need to do these calculations because the ^{14}C of the CO_2 moiety is more depleted (e.g. older) than for the DIC in toto.

In the paper, we stated specifically that we measured the age of the dissolved CO_2 using the methods described on Lines 298-301: "On a flow-through vacuum line, the headspace CH_4 and CO_2 were separated, combusted and purified, and graphitized by the sealed tube Zn reduction method¹ then measured for radiocarbon (^{14}C) on a compact accelerator mass spectrometer (AMS, National Electrostatics Corp.)"

Figures. There are many more figures than there are figure legends and so I am confused here

32125 Bio Station Lane

Polson, Montana, U.S.A. 59860-6815

Phone (406) 982-3301

Fax (406) 982-3201

<http://flbs.umt.edu>

<http://flbs.umt.edu/webcams>

flbs@flbs.umt.edu

about which figures the authors intend to be in the paper. This is messy and makes the paper difficult to fully evaluate.

We are unsure what the issue was here because the figures and legends were all present. Nonetheless, we have clarified the paper by switching the second figure with previously supplemental figures so that more of the data is visible in the main text rather than the extended data. We have also removed any redundant references to Extended Data figures, so each figure and table is only referred to once in the main text. Unfortunately the length limits prohibit incorporating more of the figures into the main text.

Reviewer #2 (Remarks to the Author):

This is a very interesting study tackling two timely issues in freshwater science: methane as a carbon source and the age of carbon entering aquatic food webs. You present quite compelling data to demonstrate that larval biomass of a number of stonefly species inhabiting flood plain aquifers is made up partially of methane-derived carbon (MDC) and that part of that carbon is ancient. These are important findings that will be of interest to others in the field and potentially to ecologists more widely. However, although the findings of the study certainly extend current knowledge of both the two issues, both of which have been attracting publications in recent years, I think you generally rather overstate the novelty of your work. The most novel aspect is how ancient carbon may be transferred via biogenic methane and MOB to consumers in this floodplain system.

We disagree – the most novel part, which we have revised the paper throughout to better express, is that we show the incorporation of millennial aged to ancient methane derived carbon in the biomass of consumers. We do not focus on the process by which this occurs, but infer potential and likely pathways.

Unfortunately the data cannot resolve whether the methane itself is "ancient" or if it is "modern" methane produced

biogenically from "ancient" organic matter (no data are provided on DOM from which any methane would likely be produced given that you reasonably convincingly discount the alternative of methane production from ancient CO₂) and this does weaken the impact and significance of the study.

32125 Bio Station Lane

Polson, Montana, U.S.A. 59860-6815

Phone (406) 982-3301

Fax (406) 982-3201

<http://flbs.umt.edu>

<http://flbs.umt.edu/webcams>

flbs@flbs.umt.edu

We do not agree the significance is weakened because we show the incorporation of aged-ancient carbon in the form of methane is contributing up to a majority of the carbon present in a very high proportion of consumer biomass, which certainly is a novel finding.

Moreover, the pathway via MOB is inferred and, while I do not really doubt it, you have not actually demonstrated the presence of MOB in the diets of the invertebrates or even in the system at all, which again is a weakness. Overall the approach and methodology are valid, but some data are lacking (isotope values for DOM and other invertebrates in the system; direct evidence for presence of MOB) that would have been very helpful and could have provided greater insight into the mechanisms/pathways involved.

We agree that these data would be helpful in assembling a complete picture of the aquifer food web and the mechanisms for incorporation of methane derived carbon (MDC) into stonefly biomass. However, as we stated in our responses to the previous two comments, this was not the objective of the paper. Whether or not the inferred MOB pathway is indeed correct, methane derived carbon is a significant proportion of stonefly biomass. In fact, if we did not assume an MOB link as used in the 'conservative' estimates of MDC contribution, the calculated MDC contribution would be consistently higher, as expressed by the 'average' estimates of MDC contribution (e.g. Table 1). Furthermore, previous papers that have demonstrated MDC in consumer biomass have used the same stable isotope methodology to show incorporation of MDC in biomass²⁻⁴, and also infer stable isotope values resulting from fractionation by MOB².

We have clarified the paper to acknowledge both of these points on lines 213-217 as follows:

“Because methanogenesis occurs mainly in anoxic environments and MOB flourish in opposing gradients of methane and oxygen⁵, it is likely that stoneflies were directly or indirectly consuming resources produced at these interfaces (i.e. either grazing or consuming via an intermediate trophic link)”

So far as I can judge the statistical approaches are sound, and the use of the Bayesian models allows uncertainties to be incorporated. The conclusions are generally valid and robust (aside from

the weaknesses already referred to), but the language needs to be tightened to make the inferences/conclusions more precise. The paper is reasonably well constructed, but was not always easy to follow. One problem is the frequent references in the text to tables and figures that are included as Extended Data; I think several of these items deserve to be within the main body of the paper and immediately available to readers rather than presented as supplementary

32125 Bio Station Lane

Polson, Montana, U.S.A. 59860-6815

Phone (406) 982-3301

Fax (406) 982-3201

<http://flbs.umt.edu>

<http://flbs.umt.edu/webcams>

flbs@flbs.umt.edu

information. Conversely, Figures 1 and 2 were of negligible value in understanding the text and could be deleted or at least moved to the Extended Data category. For a reader like me not familiar with these floodplain aquifer systems and their hydrology, a schematic diagram might be very helpful to illustrate the hydrologic pathways, the habitat/location of the animals and the potential pathways for methane and organic matter input into the system.

We have clarified this by switching the second figure with the third supplemental figure so that more of the data are visible in the main text rather than the extended data. We kept Figure 1 in the main paper because it is crucial for the reader to see the locations of the floodplains, their expansiveness, their pristine nature, and the heterogeneity of the aquifer sediments that likely mediates the redox gradients that allow methane production and consumption.

We have also removed any redundant references to Extended Data figures, so each figure and table is only referred to once in the main text. Unfortunately the length limits prohibit incorporating more of the figures into the main text.

Overall, although I enjoyed reading the paper and found the findings of considerable interest, I am afraid there are too many shortcomings and gaps in the narrative for me to recommend publication in Nature Communications. Some of these could certainly be addressed in a revision, but others are more intractable and at the very least would require additional data. I provide some more detailed specific comments below, listed by line number (L).

L1 In the title and throughout the text I am not happy with your use of the term "ancient methane". To me this implies that the methane itself is ancient, and this is just not proven by the ^{14}C aging. The methane could have been produced recently from ancient organic matter but still register the same ^{14}C age. I think the way you present the results in this context is sloppy and you need to tighten the expressions to avoid misleading inferences.

We have changed this throughout the paper to "ancient methane-derived carbon".

L14 I think you should say "much of the biomass carbon...is methane-derived." It is not all methane-derived, and your estimates range from as high as 66% to as low as 15% for those animals you have analysed.

Contrary to the reviewer's comment, our estimates of methane derived carbon in consumer biomass across the Nyack Floodplain ranged from 37.3% to 66.5% and from 8 to 41% at the

32125 Bio Station Lane

Polson, Montana, U.S.A. 59860-6815

Phone (406) 982-3301

Fax (406) 982-3201

<http://flbs.umt.edu>

<http://flbs.umt.edu/webcams>

flbs@flbs.umt.edu

other floodplains studied. Individual MDC in biomass estimates extended the full range of 0-100%.

According to formatting guidelines for Nature Communications, we cannot put statistics in the article summary. Therefore, to better express the range of methane-derived carbon contributions to consumer biomass, we have changed lines 13-15 to: "Here we solve this long standing problem by showing that up to a majority of the biomass carbon composition of these top consumers in four floodplain aquifers of Montana and Washington is methane-derived."

L15-17 Here you cannot say anything about the age of the methane - it is the age of the carbon in the methane. Also I think you could say "apparently mostly derived from biogenic sources although some thermogenic contribution cannot be excluded" to better reflect your data.

Okay, we changed L15-17 to the following: "The methane carbon ranged in age from modern to up to >50,000 years old and was mostly derived from biogenic sources, although a thermogenic contribution could not be excluded."

L18 What does "the most extensive methane-dependent freshwater ecosystem ever described" mean? Seems like hyperbole!

Okay, we changed L17-22 as follows: "This the first report of a freshwater ecosystem to contain secondary consumers with biomass comprised of ancient methane-derived carbon, and it documents one of the most expansive ecosystems to contain a majority of site-wide biomass comprised of methane-derived carbon. These findings thereby transform contemporary understanding of basal resources supporting riverine productivity."

L24 This opening sentence is a bit odd, as it is not normal practice to mention journal titles when citing literature. Seems like journal name dropping! Also, as the papers are now several decades old it is enough to say they "revolutionized" rather than "have revolutionized".

This paper is a follow-up to those landmark studies, which were published in these journals because of their broad significance. Given their significance and the degree to which this paper

32125 Bio Station Lane

Polson, Montana, U.S.A. 59860-6815

Phone (406) 982-3301

Fax (406) 982-3201

<http://flbs.umt.edu>

<http://flbs.umt.edu/webcams>

flbs@flbs.umt.edu

updates these findings, we considered those papers an appropriate context. Nonetheless, we removed the journal names as the journal names do not add to the paper.

L43 You describe the stoneflies as top consumers but in ED Fig 1 they are categorised as grazers/omnivores, so this perhaps warrants some explanation as to why there are no predatory top consumers in this ecosystem.

This issue comes up twice for reviewer 2: here and in line 133. We explain these classifications and why it is clear that the stoneflies are the top consumers in the text. This information is found from L135 to L139 as follows: “These stoneflies were by far the largest-bodied organisms living in the aquifers. Mature nymphs of *P. frontalis* and *K. perdita* are 2.5 - 3 cm long, and have mouth parts typically associated with carnivory (elongate mandibles with sharp apical teeth); whereas the *Isocapnia* spp are smaller, 1-2 cm long and have mouthparts characteristic of grazers (short mandibles with short and stout apical teeth). However, guts of all species contained amorphous, particulate organic-matter (POM), especially in early instars, and we concluded that in these dark, organic-carbon-limited aquifers, these large consumers eat whatever organic matter they encounter.”

L43 The legend to ED Fig 1 states that 17 Plecopteran species occur in the Nyack floodplain, but the figure only appears to indicate 2 species. The legend then states that 5 species commonly occurred in the well samples and these are listed in Table 1. Why do these numbers not tally?

This was a mislabeled figure and the figure has been changed to show that 17 species are only occasionally found in the aquifer (occasional hyporheos). While these 17 Plecopteran species have been identified in the aquifer, only the five highlighted species are abundant. The other species are classified as occasional hyporheos, rather than amphibionts that spend their entire larval stages in the aquifer.

We have changed the caption for ED Figure 1 to say: “Seventeen are Plecoptera, but only 5 commonly occurred in well samples (Table 1); these five species never occur in the river channel, spending the entire larval life history in the aquifer. Gibert et al. (10), described this novel life history strategy as amphibitic – hatching and growth to larval maturity occur in the aquifer, while adult emergence, mating and egg deposition are focused in the river channel.”

32125 Bio Station Lane

Polson, Montana, U.S.A. 59860-6815

Phone (406) 982-3301

Fax (406) 982-3201

<http://flbs.umt.edu>

<http://flbs.umt.edu/webcams>

flbs@flbs.umt.edu

L54 You mention a paucity of labile OC, but what about total OC? Methane can be produced biogenically from DOC that would not be considered labile.

Okay, we have now clarified the text to express both the paucity of labile DOC and the low concentrations of total DOC. Please see L54 to 56 as follows: “Overall, the aquifer is well oxygenated because oxygen diffuses from the vadose zone of the floodplain¹⁰, dissolved organic carbon concentrations are consistently less than 2 mg/L and microbial productivity is ultra-limited by paucity of labile organic carbon”

L133 "but the Isocapnia spp are thought to be grazers" - thought by whom?

As stated above in our response to the reviewer’s comment on L43, we have explained this in L135 to L139 as follows: “These stoneflies were by far the largest-bodied organisms living in the aquifers. Mature nymphs of *P. frontalis* and *K. perdita* are 2.5 - 3 cm long, and have mouth parts typically associated with carnivory (elongate mandibles with sharp apical teeth); whereas the *Isocapnia* spp are smaller, 1-2 cm long and have mouthparts characteristic of grazers (short mandibles with short and stout apical teeth) ²¹. However, guts of all species contained amorphous, particulate organic-matter (POM), especially in early instars, and we concluded that in these dark, organic-carbon-limited aquifers, these large consumers eat whatever organic matter they encounter.”

L135 Yes, it indicates a likely role for MOB but does not directly prove it, and cannot resolve whether the stoneflies are using MOB directly or indirectly by an additional food web link.

We clarified this by explaining that stoneflies are consuming methane-derived carbon, that could be in the form of MOB or an indirect food web link. We changed L 142-143 as follows: “This indicated that the stoneflies were consuming methane-derived carbon , rather than deriving low $\delta^{13}C$ values from a symbiosis with methanogenic microbes.”

L197 To reiterate, it is the carbon in the methane that represents different ages, not necessarily the methane itself.

32125 Bio Station Lane

Polson, Montana, U.S.A. 59860-6815

Phone (406) 982-3301

Fax (406) 982-3201

<http://flbs.umt.edu>

<http://flbs.umt.edu/webcams>

flbs@flbs.umt.edu

We have changed all mentions of methane carbon contributions in the paper to ‘methane-derived carbon’.

L202 on. This inference may very well be correct, but it is really entirely speculative. You have not directly demonstrated the presence of MOB in the system, or even demonstrated that methane oxidation is occurring. If stoneflies directly and preferentially graze the MOB how do they do this? Your $d^{13}C$ values for biofilm indicate that if MOB are present in the biofilm they must be a tiny proportion (do you have any microscopic or DNA evidence regarding the composition of the biofilm?), so it is difficult to see how stoneflies could preferentially graze them, especially those species with mouthparts "typically associated with carnivory" (L132). So the MOB are presumably located elsewhere, but then how might the stoneflies access them? Or is it possible that the stoneflies use them indirectly by consuming an intermediate trophic link? Here it would have been really useful to have $d^{13}C$ data for other taxa from those shown in ED Fig 1 to see if any of those also show low $d^{13}C$ values indicative of MOB incorporation. It would also have been very valuable to have some independent corroboration of MOB presence and use (directly or indirectly) by the stoneflies; for example, fatty acid analysis of stonefly tissue could reveal the presence of fatty acids that serve as specific biomarkers for MOB and this would unambiguously confirm both the presence of MOB in the system and a role for them in stonefly nutrition.

Because we show the presence of methane-derived carbon in stonefly biomass, we considered it important to discuss the most likely pathway for that to occur. However, we do not mean to claim to have proven how it occurs, nor is that important for showing that MDC is definitely in the biomass of consumers.

So, we have revised to the text in L212 to 214 as follows:

“Because methanogenesis occurs mainly in anoxic environments and MOB flourish in opposing gradients of methane and oxygen, it is likely that stoneflies were directly or indirectly consuming resources produced at these interfaces.”

L210 Surely methane at the very low methane concentrations recorded does not represent a "potential water contaminant".

Okay, we have removed the mention of a potential water contaminant and changed L218-220 as follows: “Furthermore, because the amphibitic stoneflies emerge from the river as flying or crawling adults, they are exporters of labile organic carbon from the aquifer to the floodplain

32125 Bio Station Lane

Polson, Montana, U.S.A. 59860-6815

Phone (406) 982-3301

Fax (406) 982-3201

<http://flbs.umt.edu>

<http://flbs.umt.edu/webcams>

flbs@flbs.umt.edu

surface as well as top consumers in a food web that clearly sequesters methane, a powerful greenhouse gas.”

L211-215 I really think you overstate the case here in several regards.

L215-225 I think you underplay previous work. The study by Caraco et al showed a role of ancient OM in the Hudson River, and ancient OM could contribute to your system via methanogenesis and then MOB. It is not true that all previous studies have been site-specific. Reference 26 (Shelley et al) assessed methanotrophy in relation to photosynthetic PP across a wide range of rivers in southern England; it is true that that paper did not measure methane contribution to consumers, but the implications are clear.

We refer to studies that explicitly show aged to ancient MDC in a higher trophic level and run the analysis across ecosystems. We have revised Lines 227-229 as follows:

“This is the first report of a methane-derived carbon contribution to top consumer species across multiple river ecosystems, and the first report of an ancient methane subsidy to any freshwater consumers.”

It is not fair to imply that lake studies of methane contributions only relate to north-temperate lakes; there have been reports from Arctic and tropical lakes, and Jones et al (2008, Ecology) presented data from almost 100 lakes with a global distribution. Besides the "few midge species" that have attracted most attention, there are reports of several other benthic taxa that appear to contain some methane-derived carbon based on their low $d^{13}C$, and also several reports concerning crustacean zooplankton with $d^{13}C$ values lower than those you report for stoneflies, and even reports showing transfer of MDC up to fish.

In response to this comment, we have revised the discussion emphasize the novelty of this study in terms of being the first report of MDC in consumer biomass across multiple river ecosystems and the first report of ancient methane-derived carbon in consumer biomass in any freshwater ecosystem. Therefore, our review of lake methane-derived carbon food webs is no longer relevant here. The entire revised paragraph is as follows in L227-238:

“This is the first report of a methane-derived carbon contribution to top consumer species across multiple river ecosystems, and the first report of an ancient methane subsidy to any freshwater

32125 Bio Station Lane

Polson, Montana, U.S.A. 59860-6815

Phone (406) 982-3301

Fax (406) 982-3201

<http://flbs.umt.edu>

<http://flbs.umt.edu/webcams>

flbs@flbs.umt.edu

consumers. While methane cycling and aged carbon have each been studied in rivers, the few published studies that document a river food web methane subsidy are site-specific. For example, Caraco et al. documented an ancient carbon subsidy to zooplankton in the Hudson River Estuary, Kohzu et al. showed that some macroinvertebrate production was fueled by biogenic methane produced from detritus in backwater pools, and Trimmer et al. showed that caddis fly species derived up to 30% of their biomass from methanogenic methane in the River Lambourn. None of these studies reported that methane-derived carbon was ancient. Additionally, the aquifers that we studied are dark. Therefore, the $\delta^{13}\text{C}$ depletion in stonefly biomass could not have occurred from fractionation of isotopically light CO_2 during photosynthesis, as has been found in previous studies.”

L 231 I don't think you can say the stoneflies are "methane dependent". They may use MDC, in some instances to quite a large extent, but methane-dependence surely implies that they could not manage without methane, and I do not think that is really the case.

We simply disagree. These systems are ubiquitous and ultra carbon limited. The stoneflies could not exist without a methane carbon subsidy.

Nonetheless, we have removed this claim from the discussion.

L329 It is unfortunate that your $\delta^{13}\text{C}$ values for biofilm come from these 10 week incubations of gravel bags. I wonder how well these represent the taxonomic and isotopic composition of the long-term in situ biofilm. Note also that the isotope values of these slowly maturing large stoneflies will reflect diets over quite a long span of preceding time whereas your incubated biofilms have "current" isotopic signatures.

While the taxonomic and isotopic composition of the biofilm might vary over time, the purpose of measuring biofilm $\delta^{13}\text{C}$ signatures was to include them in an aggregate organic matter (OM) pool that represented non-methane derived carbon resources available to stoneflies. Our use of these 10-week incubation estimates was valid for several reasons: 1) our overall estimate of the aggregate OM value was close to what we would expect; 2) our biofilm incubations came from the base flow period in the aquifer, when DO dropped and methanogenesis would be the most likely to occur; 3) stoneflies were so depleted in $\delta^{13}\text{C}$ that the only possible explanation for their stable isotope values was methane-derived carbon, whether or not biofilm communities had variable $\delta^{13}\text{C}$ values over time.

32125 Bio Station Lane

Polson, Montana, U.S.A. 59860-6815

Phone (406) 982-3301

Fax (406) 982-3201

<http://flbs.umt.edu>

<http://flbs.umt.edu/webcams>

flbs@flbs.umt.edu

Our calculation of the organic matter $\delta^{13}\text{C}$ estimate was explained in the text in lines 424-431 as follows, To represent any possible contribution of organic matter to stonefly diet, we used ‘organic matter’ as a surrogate for any component of the stonefly biomass that was not methane-derived carbon. Means and standard errors of $\delta^{13}\text{C}$ values for each organic matter classification are displayed in Extended Data Table 3. Coarse particulate organic matter (CPOM) showed depletion relative to other organic matter pools because stonefly detritus was inevitably and visibly incorporated into the CPOM pools we collected via pumping. We used a stratified average of all OM pools, $-27.83 \pm 2.49 \text{‰}$ which is very close to the literature estimate of photosynthetically fixed terrestrial carbon: -28‰ .”

In regards to the second part of the reviewer’s comment, stonefly stable isotope signature turnover times are unknown, but a study of ladybird beetles showed that whole-insect biomass stable isotope values reflected dietary changes in 22 days⁶, which was a time scale much less than the ten weeks of biofilm sample incubation time. Regardless, given the validity of the organic matter $\delta^{13}\text{C}$ estimate and the year-round occurrence of stonefly $\delta^{13}\text{C}$ depletion, these turnover times would not affect overall findings of methane derived carbon in biomass.

L356, Strictly, MOB do not "prefer" the lighter isotope; rather the lighter isotope is preferentially utilised in their metabolism.

Okay, we changed this to L372 to 374 as follows: “All methane can then be consumed by methane oxidizing bacteria (MOB), which fractionate the dissolved methane by preferentially assimilating the lighter carbon isotope, leaving the residual methane enriched in the heavier isotope.”

L394-398 Some other studies have been complicated by the possibility that methane carbon could enter food webs by oxidation to heavier but still very light CO₂ which is then incorporated into algae by photosynthesis and then passed to consumers; so not by direct consumption of MOB. In fact van Duinen et al (2013, Freshwater Science) argued that this was the predominant pathway by which MDC reached invertebrates in Estonian bog pools. In your dark hyporheic system this is presumably not a possible pathway, and the data you present for CO₂ nicely confirm this. I think you should make this point more strongly in the paper as not all potential readers will necessarily appreciate it otherwise.

32125 Bio Station Lane

Polson, Montana, U.S.A. 59860-6815

Phone (406) 982-3301

Fax (406) 982-3201

<http://flbs.umt.edu>

<http://flbs.umt.edu/webcams>

flbs@flbs.umt.edu

We added this in to our discussion in L 235-238 as follows: “None of these studies reported that methane-derived carbon was ancient. Additionally, the aquifers that we studied are dark. Therefore, the $\delta^{13}\text{C}$ depletion in stonefly biomass could not have occurred from fractionation of isotopically light CO_2 during photosynthesis, as has been found in previous studies.”

L422 Specify "photosynthetically fixed TERRESTRIAL carbon" - aquatic photosynthesis can produce a much wider range of $\delta^{13}\text{C}$ values for organic matter.

Okay, we changed this to L 429-431 as follows: “We used a stratified average of all OM pools, $-27.83 \pm 2.49 \text{‰}$ which is very close to the literature estimate of photosynthetically fixed terrestrial carbon: -28‰ .”

Sincerely,

Amanda DelVecchia

1. Xu, X. *et al.* Modifying a sealed tube zinc reduction method for preparation of AMS graphite targets: reducing background and attaining high precision. *Nucl. Instrum. Methods Phys. Res. Sect. B Beam Interact. Mater. At.* **259**, 320–329 (2007).

32125 Bio Station Lane

Polson, Montana, U.S.A. 59860-6815

Phone (406) 982-3301

Fax (406) 982-3201

<http://flbs.umt.edu>

<http://flbs.umt.edu/webcams>

flbs@flbs.umt.edu

2. Trimmer, M., Hildrew, A. G., Jackson, M. C., Pretty, J. L. & Grey, J. Evidence for the role of methane-derived carbon in a free-flowing, lowland river food web. *Limnol. Oceanogr.* **54**, 1541–1547 (2009).
3. Kohzu, A. *et al.* Stream food web fueled by methane-derived carbon. *Aquat. Microb. Ecol.* **36**, 189–194 (2004).
4. Jones, R. I. & Grey, J. Biogenic methane in freshwater food webs. *Freshw. Biol.* **56**, 213–229 (2011).
5. Bussmann, I., Rahalkar, M. & Schink, B. Cultivation of methanotrophic bacteria in opposing gradients of methane and oxygen. *Fems Microbiol. Ecol.* **56**, 331–344 (2006).
6. Ostrom, P. H., Colunga-Garcia, M. & Gage, S. H. Establishing pathways of energy flow for insect predators using stable isotope ratios: field and laboratory evidence. *Oecologia* **109**, 108–113 (1996).